# Upregulation of interleukin-1β and interleukin-18 in traumatic brain injury patients and their potential as biomarkers

**Saiqa Nazeer, Ghulam Murtaza** [ID]*

Department of Zoology, University of Gujrat, Gujrat, Pakistan

* gmurtazay@yahoo.com

## Abstract

### Background

Traumatic brain injury (TBI) is diagnosed using radiological imaging and biomarker analysis, each with certain limitations. Accurate and accessible detection of TBI is essential for its treatment. Since interleukin (IL)-1β and IL-18 are key neuroinflammatory cytokines, we aim to investigate the regulation of IL-1β and IL-18 in TBI patients and assess their potential as biomarkers of TBI.

### Methods

Forty TBI and 40 healthy subjects were recruited for this study. The Glasgow coma scale (GCS) was used to assess the clinical severity of TBI. Spearman's rank-order correlation was employed to find the association between GCS scores and IL levels of TBI patients. An enzyme-linked immunosorbent assay (ELISA) with human IL-1β and IL-18 ELISA kits was conducted to measure the serum IL-1β and IL-18 levels. The area under the curve (AUC), sensitivity, and specificity were measured to assess the potential of IL-1β and IL-18 as biomarkers.

### Results

The patients had mild-to-moderate TBI (mTBI) with a mean GCS score of $11.05 \pm 1.67$. A negative correlation was found between GCS scores and IL levels, while IL-1 β and IL-18 were positively correlated with each other. IL-1β levels in the TBI and healthy subjects were $1882.19 \pm 796.72$ pg/mL and $1473.50 \pm 333.045$ pg/mL, respectively, with a significant difference ($p = 0.001$). Moreover, the IL-18 levels in TBI subjects ($25.92 \pm 14.81$ ng/L) were also significantly higher ($p < 0.001$) compared to those of the control group subjects ($19.14 \pm 2.99$ ng/L). The IL-1β exhibited an AUC of 0.72 with 72.5% sensitivity and 70% specificity, at a cut-off point of 1527.25 (pg/mL), while IL-18 had an AUC of 0.78 with 77.5% sensitivity and 72.7% specificity, at a cut-off point of 19.645 (ng/L).

**Data availability statement:** All relevant data are within the paper and its Supporting information files.

**Funding:** The author(s) received no specific funding for this work.

**Competing interests:** The authors have declared that no competing interests exist.

## Conclusions

IL-1β and IL-18 levels were increased remarkably in mTBI patients and exhibited fair potential as biomarkers for the diagnosis and prognosis of TBI.

## 1 Introduction

Traumatic brain injury (TBI) is a disruption of brain functioning caused by different factors such as shock, collision, penetrating head injury, traffic accidents, falls, violence, participation in military activities, and contact sports such as boxing, hockey, and football [1,2]. It is considered a major socioeconomic and health problem worldwide, having the highest incidence of all commonly known neurological disorders [3]. Severity of TBI is measured in terms of mild, moderate, and severe, which is assessed by the Glasgow coma scale (GCS) [4]. Mild TBI (mTBI) causes changes in the molecular and structural composition of the brain [5], while moderate to severe TBI, in addition to these disturbances, may lead to profound and persistent deficits in cognitive functions [6]. This disease not only affects the daily life of people but also incurs a burden on health departments, particularly in low-income countries [7].

Neuroimaging techniques such as computed tomography (CT), imaging spectroscopy, magnetic resonance imaging (MRI), and diffusion tensor imaging (DTI) are being applied to diagnose TBI. However, they cannot detect the minute brain changes [8] and are expensive, with limited availability across clinics [9]. In addition to these advanced techniques, biomarkers (substances that help in the diagnosis, progression, prevention, understanding, and treatment of different diseases and the response to therapy), such as cytokines, are also employed for the detection and monitoring of TBI, based on analysis of both cerebrospinal fluid (CSF) and serum [10–13]. Various inflammatory cytokines have been demonstrated as potential biomarkers of TBI in both CSF and serum. CSF biomarkers have limitations as they face hindrance while moving through the blood-brain barrier (BBB) and are produced in very low concentrations [14]. In comparison, serum biomarkers can easily be obtained from blood and analysed immediately after trauma, thus can serve as a convenient tool for disease detection and monitoring [15]. However, many studies investigating molecules as biomarkers have not reported their area under the curve (AUC), sensitivity, and specificity values, which are crucial to determine their authenticity as biomarkers [16].

TBI shows biphasic pathology in which the initial traumatic effects cause persistent inflammation, and the innate immune system is activated [17,18]. The inflammatory process induces cytokine secretion [19], including interleukins (ILs) [20]. Upregulation of ILs following TBI has also been demonstrated in both human [21] and murine brains [22]. The immune response is influenced not only by environmental factors but also by the genetic makeup of the organisms. Consequently, responses to different diseases may vary among racial and ethnic groups in certain populations and regions of the world [23]. The pathology of TBI is highly complex and involves many mechanisms, such as oxidative stress, neuronal damage, neuronal excitotoxicity, and

inflammatory responses [24]. Although an extensive literature is present on biomarkers to assess TBI, only a few, such as the Food and Drug Administration (FDA)-approved combined measurement of glial fibrillary acidic protein (GFAP) and ubiquitin C-terminal hydrolase-L1 (UCH-L1), are currently employed for the early diagnosis of intracranial lesions following closed head injury [25]. Despite this advancement, there is continued scientific effort to identify and validate additional bio-markers linked to neuroinflammation to improve prognostic assessment, rather than serving solely as diagnostic tools [26].

Based on these considerations, we aimed to investigate the link between the key inflammatory cytokines IL-1β and IL-18 and TBI in the Pakistani population by using serum to measure their levels. Moreover, their diagnostic and prognostic potential was assessed by evaluating their AUC, sensitivity, and specificity values. In this work, IL-1β and IL-18 are selected, as they are major upstream regulators of TBI-induced inflammation and secondary brain injury [27]. Unlike GFAP and UCH-L1, which are primarily upregulated as a result of neuronal and glial cell damage [28,29], these cytokines provide insight into the activation of the inflammatory pathway, aligning with the main focus of our study.

## 2 Methods

### 2.1 Selection of subjects

In this cross-sectional study, patients having road accident-caused TBI (N = 40) were voluntarily recruited from the Emergency Department of Major Aziz Bhatti Shaheed Teaching Hospital, Gujrat, Pakistan, between July 2022 and June 2023. The control group, comprising healthy individuals (N = 40), was recruited from the same city. Informed written permission was acquired from all subjects for their inclusion in this study, and a written questionnaire was used to record their demographic, social, economic, and health characteristics. The TBI and control groups were matched for demographic variables, including age, gender, marital status, education, and monthly income, to avoid confounding effects. Patients with pre-existing inflammatory, autoimmune, infectious, or neurological disorders, as well as those with a prior history of TBI or concussion, were excluded from this study to avoid confounding effects on circulating cytokine levels. Inclusion criteria were age between 18 and 65 years, trauma duration less than 24 hours, and consciousness. The GCS score was measured out of 15. Scores of 13–14, 9–12, and 3–8 denote mild, moderate, and severe TBI, respectively [30].

### 2.2 Serum extraction and enzyme-linked immunosorbent assay (ELISA)

For the extraction of serum, 5 mL of venous blood was drawn from TBI patients at the time of hospital admission and from healthy subjects at recruitment under resting conditions. Blood samples were kept for clotting at room temperature for one hour. Later, centrifugation of the blood was done at 2000−3000 RPM for 20 minutes. The supernatant (serum) of the centrifuged material was moved to a separate tube and kept at −80°C until further processing.

ELISA kits (Bioassay Technology Laboratory, Birmingham B15 3LB, United Kingdom) were used for the detection and quantification of ILs. The ELISA plates used were pre-coated with human IL-1β antibody and human IL-18 antibody, respectively. Fifty microliters of the standard solution for both ILs was added to the standard wells separately. Forty microliters of serum collected from TBI and healthy subjects containing IL-1β and IL-18 (antigens) were added to the respective wells, which bound to the antibodies. Then, added 10 μL of biotinylated human anti-IL-1β antibody and biotinylated human anti-IL-18 antibody separately to the respective wells. Fifty microliters of Streptavidin-HRP was added for both ILs and mixed thoroughly [31]. Plates were sealed and incubated for 60 minutes at 37°C. Then, after removing the sealer from the plates, the wash buffer was used to wash the plates. Decanted each well and washed 5 times with the wash buffer. Paper towels were used to blot the plates. Later, 50 μL of the substrate solution A was added to each well, after which 50 μL of substrate solution B was added to each well for both ILs. The plates were incubated for 10 minutes at 37°C under dark conditions. Then, 50 μL of the acidic stop solution was added to each well for both ILs. For both IL-1β and IL-18, the blue colour was changed to yellow immediately. Optical density (OD) was measured immediately with a microplate reader at 450 nm (model MB-580, Heales, Hitchin, UK) [32]. A standard curve was obtained by plotting the mean OD for each

 

standard on the Y-axis versus concentrations on the X-axis. Computer-based curve-fitting software was employed to carry out calculations, with the best-fit line obtained by the regression analysis.

## 2.3 Ethical statement

All procedures were carried out following the standards of the Institutional Review Board (IRB), University of Gujrat, which approved the study before its commencement (Ref: UOG/ORIC/2022/204). Furthermore, the research work was carried out according to the Declaration of Helsinki.

## 2.4 Statistical analysis

The statistical package for the social sciences (SPSS, IBM, version 21) was used to analyse the results. Data were presented as mean±standard deviation (SD). The Kolmogorov-Smirnov test exhibited a non-normal distribution of data ($p<0.05$). Thus, a non-parametric Mann-Whitney U-test was employed for the comparison of IL levels between two independent groups; TBI patients and normal individuals. Statistical significance was set at $p \leq 0.05$. The discriminatory ability of ILs as biomarkers was measured based on the AUC analysis. Higher values of AUC correspond to stronger predictive accuracy. Cut-off points were selected closer to the value of AUC, and sensitivity and specificity were measured. As the data were not normally distributed, the association between GCS scores and IL levels was evaluated by applying Spearman's rank-order correlation.

# 3 Results

## 3.1 Demographic characteristics of TBI patients and normal individuals

Socio-demographic characteristics, including age (18–65 years), education (10–14 years), and monthly income (350–700 US Dollars), were the same between TBI and healthy subjects. TBI patients and healthy individuals had mean ages of 40.34±11.18 years (mean±standard deviation, SD) and 40.45±11.23 years, respectively. Of 40 TBI and healthy subjects, 62.5% and 55% were males, and 37.5% and 45% were females, respectively. Likewise, 25% and 32.5% were single, and 75% and 67.5% were married, respectively (Table 1).

## 3.2 Correlation between GCS score and IL levels

A significant negative correlation was found between GCS score and both IL-1β ($\rho=-0.430$, $p=0.006$) and IL-18 ($\rho=-0.410$, $p=0.009$), demonstrating that lower GCS scores were moderately associated with higher levels of cytokines. Moreover, a moderate positive correlation was observed between IL-1β and IL-18 levels ($\rho=0.436$, $p=0.005$), showing

**Table 1. Demographics of TBI patients and normal individuals.**

| Variables | | TBI patients N=40 | Normal individuals N=40 |
|---|---|---|---|
| Age | Years (mean±SD) | 40.34±11.18 | 40.45±11.23 |
| Gender | Male, N (%) | 25 (62.5) | 22 (55) |
| | Female, N (%) | 15 (37.5) | 18 (45) |
| Marital status | Single, N (%) | 10 (25) | 13 (32.5) |
| | Married, N (%) | 30 (75) | 27 (67.5) |
| Education | Years | 10-14 | 10-14 |
| Monthly income | US Dollars | 350-700 | 350-700 |

Abbreviations: TBI, traumatic brain injury; SD, standard deviation.

that both cytokines increased together following TBI (Table 2). The scatterplots in Fig 1A–C demonstrate these correlations. Most of the TBI patients exhibited IL-1β and IL-18 levels within or slightly above the range observed in healthy controls (IL-1β: 786.80–2250.70 pg/mL; IL-18: 13.78–27.91 ng/L). However, some TBI patients showed elevated cytokine levels, with IL-1β peaking at 5037.90 pg/mL and IL-18 at 85.69 ng/L. The GCS scores of TBI patients ranged from 6 to 14, with a mean of 11.05 ± 1.68, indicating that the cohort predominantly consisted of patients with mild-to-moderate TBI.

### 3.3 Cytokines elevation in TBI patients and their gender-based comparison

Serum samples of TBI patients were analyzed to determine the levels of IL-1β and IL-18. The levels of IL-1β protein in TBI and healthy subjects were 1882.19 ± 796.72 pg/mL and 1473.50 ± 333.045 pg/mL, respectively. This difference was significant (p = 0.001) between the two groups (Fig 2A). Similarly, the level of IL-18 in the serum of TBI patients (25.92 ± 14.81 ng/L) was significantly higher (p < 0.001) compared to its level in the serum of control group subjects (19.14 ± 2.99 ng/L) (Fig 2B). Levels of cytokines were also analyzed to assess their distribution between male and female participants. IL-1β levels were not significantly different between males and females of normal (p = 0.15) and TBI groups (p = 0.352). On the contrary, in the normal group, females had slightly but significantly higher levels of IL-18 than males (19.93 ± 2.64 vs. 18.67 ± 3.14 ng/L; p = 0.043). In TBI subjects, this difference was significantly higher (p = 0.012) in females (28.91 ± 18.03) as compared to males (23.48 ± 11.41 ng/L) (Table 3, Fig 3).

### 3.4 ROC curve and predictive power analysis of IL-1β and IL-18

To evaluate the potential of ILs as reliable biomarkers, ROC curves for IL-1β and IL-18 were constructed. The ROC curve graphically represents sensitivity (true positive rate, TPR) plotted versus 1-specificity (false positive rate, FPR), calculated for all potential cut-off values between cases and controls. For IL-1β, the AUC value was 0.72 (95% CI: 61%−84%, p = 0.001). At the cut-off point of 1527.25 pg/mL, 72.5% sensitivity and 70% specificity were calculated. Moreover, IL-18 exhibited an AUC value of 0.78 (95% CI: 67%-88%, p < 0.001) with 77.5% sensitivity and 72.7% specificity at a cut-off point of 19.645 ng/L (Table 4; Fig 4). Since IL-1β and IL-18 have AUC values > 0.7, their potential as reliable biomarkers for detecting TBI is fair (acceptable).

## 4 Discussion

Injury to the brain causes a sudden onset of an inflammatory response. Several pro-inflammatory biomarkers, including tumor necrosis factor (TNF)-α, apoptosis-associated speck-like protein containing a caspase recruitment domain (ASC), IL-1, IL-6, and IL-18, are described as clinically relevant biomarkers that may assist in the diagnosis and prognosis of brain trauma [33,34]. In this study, we observed high levels of cytokines (IL-1β and IL-18) in serum samples collected from TBI patients from the Pakistani population. The research analyzed IL-1β and IL-18 as biomarkers of TBI by comparing

**Table 2. Spearman's rank-order correlation coefficients (ρ) for correlations between GCS score and IL-1β and IL-18 levels in TBI patients.**

| Association | Spearman's ρ (rho) | P-value (two-tailed) |
| --- | --- | --- |
| GCS versus IL-1β | −0.430 | 0.006 |
| GCS versus IL-18 | −0.410 | 0.009 |
| IL-1β versus IL-18 | 0.436 | 0.005 |

Positive and negative ρ values indicate positive and negative correlations, respectively. All correlations were statistically significant (p ≤ 0.05).

Abbreviations: GCS, Glasgow coma scale; IL, interleukin; TBI, traumatic brain injury.

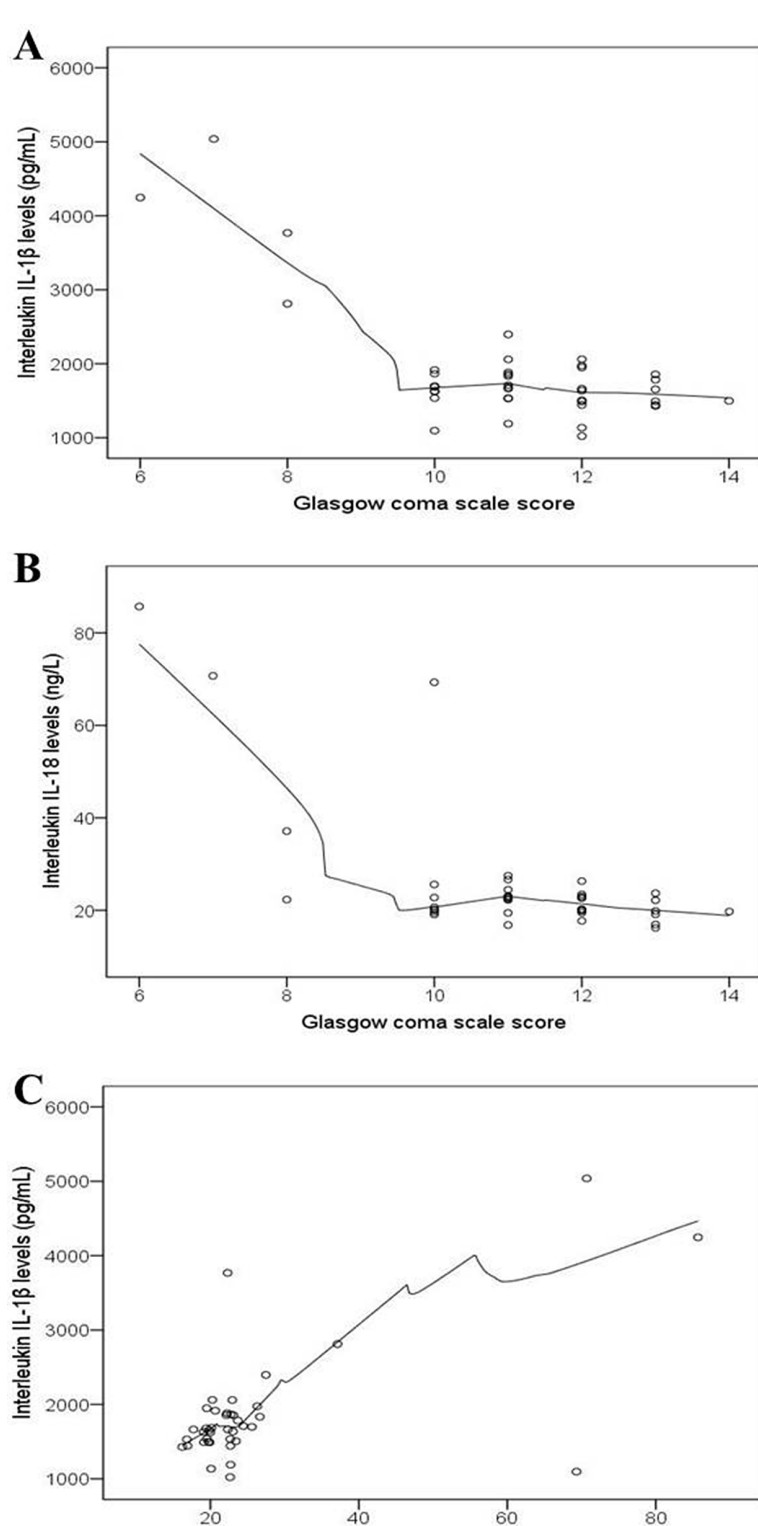

**Fig 1. Scatterplots exhibiting association between Glasgow coma scale (GCS) scores and serum interleukin (IL) levels.** GCS scores were significantly negatively correlated (Spearman's $\rho = -0.43$, $p = 0.006$) with IL-1β (A) and IL-18 (B) in TBI patients.

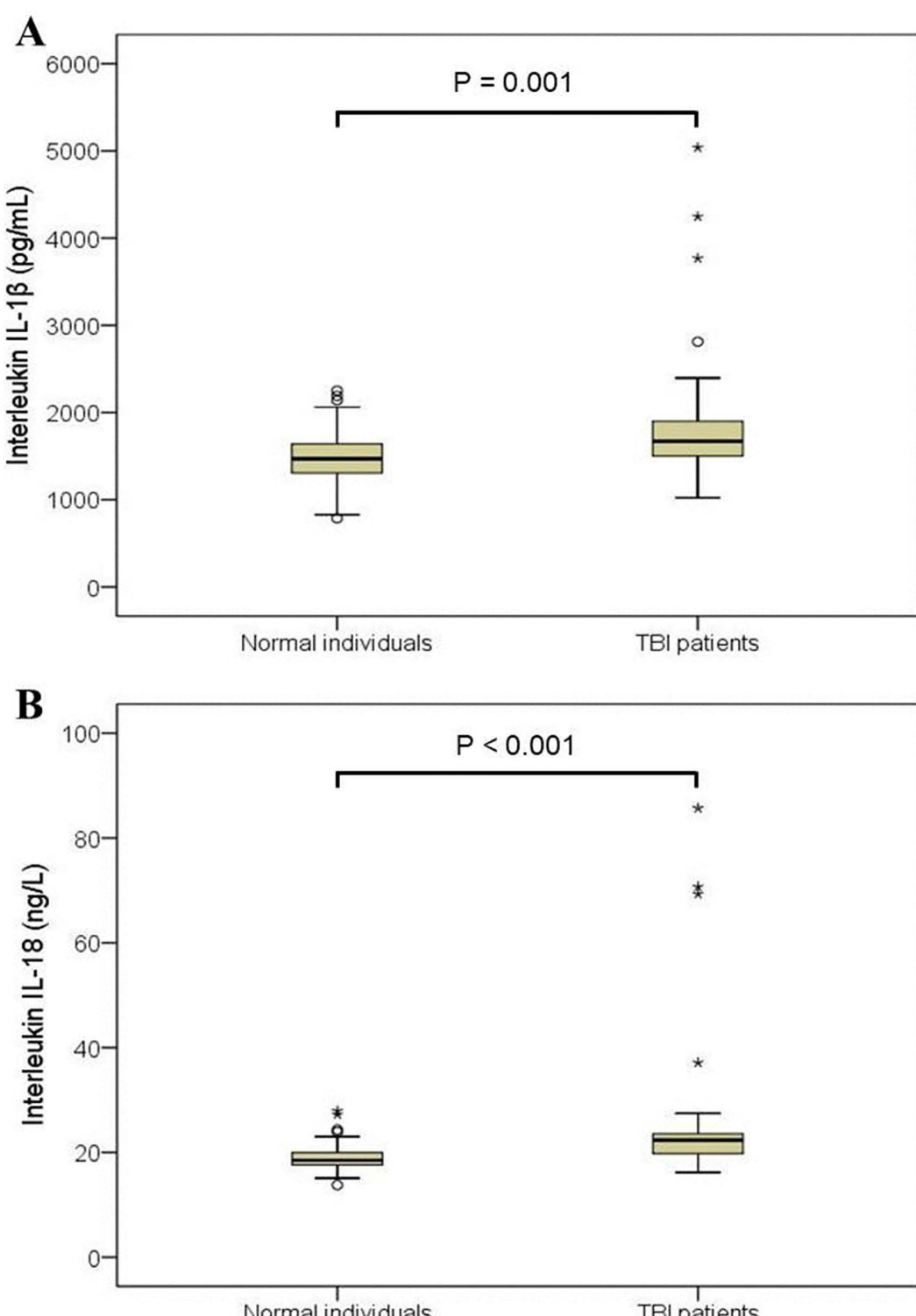

**Fig 2. Box plots exhibiting comparative analysis of interleukin (IL) levels in serum of traumatic brain injury (TBI) patients (*n*=40) and normal individuals (*n*=40), as assessed by enzyme-linked immunosorbent assay (ELISA).** Levels of (A) IL-1β and (B) IL-18 are higher in TBI patients compared with normal individuals. Each box indicates interquartile range (IQR), the horizontal line inside the box exhibits the median, whiskers depict variability outside the upper and lower quartiles, circles (°) denote mild outliers (values between 1.5 and 3 times the IQR), and asterisks (*) indicate extreme outliers (values exceeding 3 times the IQR). Statistical significance was assessed using the Mann–Whitney U-test.

**Table 3. Gender-based comparison of cytokines in normal and TBI subjects.**

| Cytokine | Group | Male (Mean ± SD) | Female (Mean ± SD) | P-value |
|---|---|---|---|---|
| IL-1β | Normal | 1538.05 ± 279.87 | 1365.92 ± 393.54 | 0.15 |
| | TBI | 1907.47 ± 760.48 | 1851.29 ± 860.21 | 0.352 |
| IL-18 | Normal | 18.67 ± 3.14 | 19.93 ± 2.64 | 0.043 |
| | TBI | 23.48 ± 11.41 | 28.91 ± 18.03 | 0.012 |

Abbreviations: IL, interleukin; TBI, traumatic brain injury; SD, standard deviation.

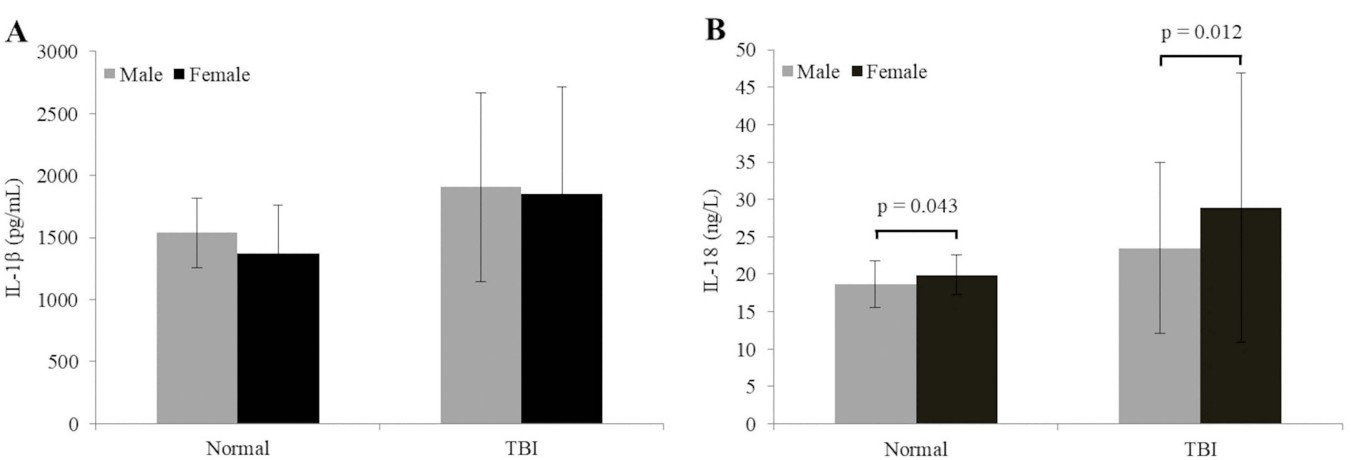

**Fig 3. Gender-based comparison of IL-1β (A) and IL-18 (B) levels in normal and TBI participants.** Data are presented as mean ± standard deviation. Statistical significance was assessed using the Mann–Whitney U test. IL-1β levels did not differ significantly between sexes in either group (p > 0.05), whereas IL-18 levels were significantly higher in females than in males in both the normal and TBI groups.

**Table 4. Diagnostic performance of interleukins for TBI: Area under the curve, 95% confidence interval, cut-off values, sensitivity, and specificity.**

| Interleukin | Interleukin-1β | Interleukin-18 |
|---|---|---|
| Area under the curve | 0.72 | 0.78 |
| 95% confidence interval (%) | 61-84 | 67-88 |
| Cut-off values | 1527.25 (pg/mL) | 19.645 (ng/L) |
| Sensitivity (%) | 72.5 | 77.5 |
| Specificity (%) | 70 | 72.7 |

data from TBI patients with that of healthy individuals. Moreover, both groups were matched for demographic characteristics, as IL levels may vary with the age of the subjects [35].

A considerable body of literature is available in which different methods have been demonstrated for the diagnosis and clinical prognosis of TBI. For instance, CT scans are employed for the detection of hemorrhagic incidents [36]. MRI can evaluate changes at minute levels, such as bleeding and diffuse axonal injury, which are otherwise not detectable via CT scans. Magnetic resonance spectroscopy (MRS) can even detect biochemical and metabolic events happening in the brain [37], as evidenced by a previous study [38], in which cellular damage was demonstrated in white matter that looked normal. However, all these advanced techniques have limitations, including exposure to radiation, time-consuming,

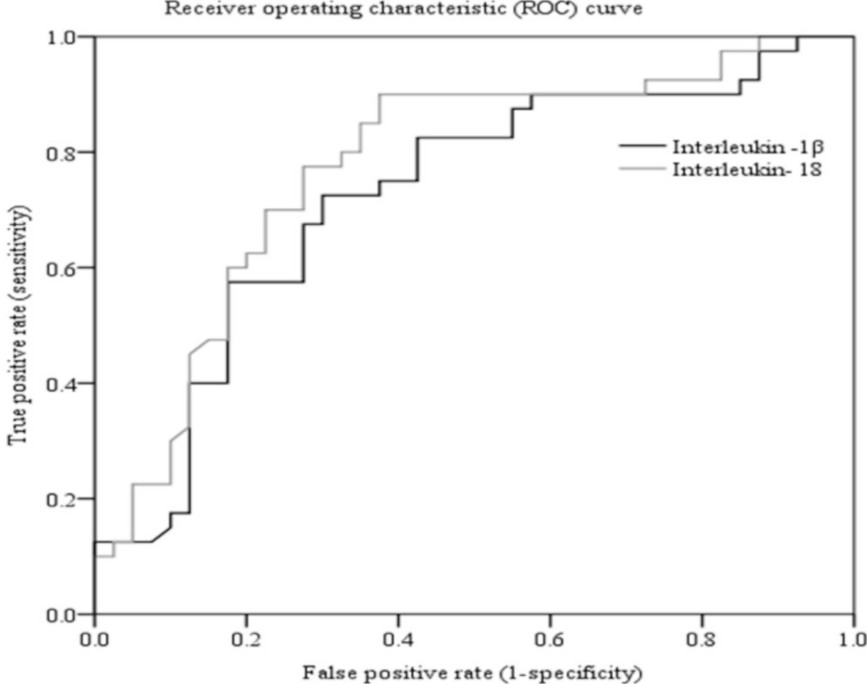

**Fig 4. The area under the receiver operating characteristic (ROC) curve (AUC) and the discriminatory ability of IL-1β and IL-18.** The black line in the figure represents IL-1β, and the grey line denotes IL-18. The Y-axis shows the ability of ILs to correctly detect TBI patients, and the X-axis represents the proportion of healthy individuals wrongly classified as TBI. The higher AUC value of IL-18 (0.78) than that of IL-1β (0.72) demonstrates its better diagnostic power as a biomarker.

expensive, and not available in all clinical settings [39]. CSF biomarkers are also utilized for the diagnosis of brain trauma, but also offer limitations of difficulty in moving through the BBB and production in small quantities [14]. In our investigation, we have chosen serum for certain reasons, as it is easily available, easily collected, time-saving, cost-effective, and free of clotting factors. Moreover, in low-income countries like Pakistan, the aforementioned advanced techniques are not available in all health care systems. Thus, application of serum to diagnose TBI at the time of hospital admission can help clinicians early detect TBI and monitor its clinical prognosis.

AUC values demonstrate the reliability of biomarkers as a diagnostic tool. In this regard, AUC values of biomarkers ranging from 0.9–1.0, 0.8–0.9, 0.7–0.8, 0.6–0.7, and 0.5–0.6 classify them as excellent, good, fair, poor, and fail, respectively [40]. In a previous study carried out with serum samples [41], the AUC value of IL-1β was reported to be lower (0.63), with 70% sensitivity and 51% specificity. Kerr et al. demonstrated that in CSF, the concentration of IL-1β was below the detection level and its AUC values could not be analysed [42]. In our research work, the comparatively higher AUC (0.72), sensitivity (72.5%), and specificity (70%) values of IL-1β may be attributed to differences in sample type and disease conditions [43]. In our study, IL-18 had an AUC of 0.78, with 77.5% sensitivity and 72.5% specificity. As reported previously, IL-18 in CSF had an AUC of 0.84, with a sensitivity of 80% and a specificity of 68.42%, suggesting it as a reliable biomarker of TBI [42]. Similarly, IL-18 in blood serum exhibited an AUC of 0.81, with 83% sensitivity and 74% specificity [33]. Thus, our results of IL-18 are comparable and in line with the previous studies [33,42]. Moreover, in light of our findings, the potential of IL-1β and IL-18 as biomarkers for the diagnosis and prognosis of TBI can be graded as fair.

In this study, we found a significant difference in serum ILs between TBI and normal subjects. In a previous study [44], elevated levels of different cytokines, including IL-1β, were demonstrated in human brain tissues following acute brain injury.

Peak concentrations of IL-1β were recorded within 24 hours in the CSF of severe TBI patients. However, the concentrations of IL-1β decreased gradually after that duration [45]. Another study reported high levels of IL-1β and IL-18 in CSF of aneurysmal subarachnoid hemorrhage patients and suggested them to be used to diagnose early brain injury and the progress of the treatment [46]. Similarly, levels of IL-18 protein were recorded as upregulated in CSF collected from patients of closed head injury (CHI) [22]. Thus, the above-cited literature evidences that under different conditions of TBI, levels of IL-1β and IL-18 are elevated. Our data exhibiting upregulated levels of IL-1β and IL-18 proteins in TBI patients are in concordance with the previous studies [22,44–46]. Our data exhibited relatively high values of IL-1β proteins in both TBI patients and controls, which could be attributed to the type of ELISA kit used and its detection range [47]. Moreover, in our investigation, since both patient and control samples were analysed on the same plate, the relative comparisons remain valid.

Our data demonstrate a differential post-TBI inflammatory response between male and female subjects, denoted by higher IL-1β levels in males and elevated IL-18 levels in females. Our results align with the previous research work that also demonstrates gender-specific inflammatory response following TBI. For instance, in an animal study [48], male mice exhibited a significant increase in IL-1β levels compared to females following experimental TBI. Conversely, gender-specific data on IL-18 levels in the context of TBI are sparse. However, our findings are supported by a previous study [49] in which higher serum levels of IL-8, another related inflammatory cytokine, were demonstrated in TBI females compared to males. Taken together, our data demonstrate that females may exhibit a more robust cytokine response following TBI, which highlights the importance of gender-specific considerations in TBI management and immune-targeted therapeutic strategies.

Levels of biomarkers (ILs) may vary under different conditions, such as sample type, organism type, severity and duration of injury, genetics, physical activity, and population characteristics. In serum samples of neonates with perinatal hypoxia (PNH), levels of IL-1β and IL-18 were higher than those of CSF samples [50]. In a study of more than 400 animal species, conservation and divergence of functional domains of IL-1 members were recognized with a suggestion of their diverse roles and patterns of expression among different groups of animals [51]. In mice having CHI, high levels of IL-18 proteins were detected between 4 and 24 hours and after 7 days of injury [22], whereas other cytokines, including IL-1β, were elevated within hours following CHI [52]. Similarly, a rise in the IL-1β levels was also reported in the TBI rat brain within the first 24 hours after injury [53]. The dynamic changes in IL-18 levels in response to injury in the rat brain were also evidenced previously [54]. Peake et al. demonstrated high levels of cytokines, including IL-1β, for several days after hard and prolonged exercise in humans [55]. The genetic variation and its association with the development of post-traumatic epilepsy (PTE) were investigated in 256 adults with TBI. The C/T (cytosine/thymine) heterozygous genotype was found to be associated with increased susceptibility to post-traumatic epilepsy. In the C/T genotype group, higher IL-1β CSF/serum ratios and lower serum IL-1β levels were observed [56]. In a review article, a comparison of biomarkers was performed among 7 ethnic groups. Remarkable differences in results for most of the markers were demonstrated among these groups [57]. Considering all the above-discussed literature, particularly variations of ILs in different populations, we investigated the levels of IL-1β and IL-18 in mTBI patients in the Pakistani population. Thus, our data might play an important role in defining the levels of the above-mentioned ILs in patients with accident-caused TBI and their differences from those of the healthy population.

This study also has some limitations that should be taken into account when the data are evaluated. Our data were not drawn from the general population, as they were collected from one small part of the country. The study lacks a comparison of investigated biomarkers among various ethnic groups, as only one ethnic group, Punjabi-speaking people, was recruited for this study. Moreover, the age distribution of the study sample may also have influenced the outcomes, further limiting the applicability of the findings. In addition, data were collected only at the time of admission and not at later stages, such as days and during/after recovery, which limited our ability to evaluate temporal changes or long-term outcomes. Another limitation of our study is the inability to determine the exact mean time of collection of blood, as the duration after the accident or admission to the hospital could not be recorded. This lack of exact timing of blood collection may have influenced the measurement of ILs.

## 5 Conclusions

IL-1β and IL-18 are upregulated in TBI patients injured in road accidents. Spearman's correlation exhibits links between clinical severity and IL levels. These ILs have fair potential to be employed as biomarkers for diagnosis and prognosis of mTBI. However, their levels may vary with sample type, injury type and duration, and population differences. Thus, a careful evaluation of data from TBI subjects is recommended before employing IL-1β and IL-18 as reliable biomarkers for detecting TBI.

## Supporting information

**S1 File. Raw data.**
(XLSX)

## Author contributions

**Conceptualization:** Saiqa Nazeer, Ghulam Murtaza.

**Data curation:** Saiqa Nazeer, Ghulam Murtaza.

**Formal analysis:** Saiqa Nazeer, Ghulam Murtaza.

**Investigation:** Saiqa Nazeer, Ghulam Murtaza.

**Methodology:** Saiqa Nazeer, Ghulam Murtaza.

**Project administration:** Ghulam Murtaza.

**Resources:** Ghulam Murtaza.

**Supervision:** Ghulam Murtaza.

**Validation:** Saiqa Nazeer, Ghulam Murtaza.

**Visualization:** Saiqa Nazeer, Ghulam Murtaza.

**Writing – original draft:** Saiqa Nazeer, Ghulam Murtaza.

**Writing – review & editing:** Saiqa Nazeer, Ghulam Murtaza.

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
