## [Decision Letter · Decision Letter 0]

30 Dec 2025

Dear Dr. Murtaza,

Thank you for submitting your manuscript to PLOS ONE. After careful consideration, we feel that it has merit but does not fully meet PLOS ONE’s publication criteria as it currently stands. Therefore, we invite you to submit a revised version of the manuscript that addresses the points raised during the review process.

We look forward to receiving your revised manuscript.

Kind regards,

Subhra Mohapatra, MS PhD

Academic Editor

PLOS One

Journal Requirements:

https://journals.plos.org/plosone/s/file?id=ba62/PLOSOne_formatting_sample_title_authors_affiliations.pdf ..

2. Please delete the separate files, leaving only the tables in the manuscript file.

Reviewers' comments:

Reviewer's Responses to Questions

**Comments to the Author**

1. Is the manuscript technically sound, and do the data support the conclusions?

Reviewer #1: Yes

Reviewer #2: Yes

2. Has the statistical analysis been performed appropriately and rigorously?

Reviewer #1: Yes

Reviewer #2: Yes

3. Have the authors made all data underlying the findings in their manuscript fully available?

Reviewer #1: Yes

Reviewer #2: Yes

4. Is the manuscript presented in an intelligible fashion and written in standard English?

Reviewer #1: Yes

Reviewer #2: Yes

Reviewer #1: This manuscript reports the results of a cross-sectional study comparing TBI patients with normal healthy controls. The study is technically sound, but there are some minor, fixable concerns.

There is no justification for the choice of cytokines studied.

Lines 100-105. The authors state there is no FDA-backed biomarkers for TBI. This is no longer true. Combined use of GFAP & UCH-L1 from blood is a good diagnostic tool.

Inclusion criteria are clear. However, Table 1 shows that the TBI and control groups were matched according to their demographic characteristics, but that was not stated in the methods.

The reported methods for obtaining and processing blood are standard. While blood was obtained on hospital admission from the TBI group, there is no indication when normals gave blood.

The ELISA method was routine.

The statistics were appropriate for the study design.

The figures were not included in the submission. I got the figures from their posting to BioRxiv

Tables 2 & 3 are very repetitive with text and Figures.

Line 174. “negative trend”. No trend analysis was done. Negative tendency might be a better term.

Line 175. They should define what they consider a normal range.

Line 176. The authors appropriately define what the different GCS scores mean and state that their subjects have mild TBI. However, they report a mean score of 11.05, which is indicative of moderate severity TBI. Please clarify.

Table 2. The correlation and p values for IL-1 & IL-18 are exactly the same. Is this correct?

Line 186 & 197 is p= 0.000 a typo?

The authors provide context for their results without overstating them.

Line 206. The authors make a general statement about pro-inflammatory biomarkers being used for diagnosis/prognosis. Which ones.

Line 228 type. .8-9.0 should be .8-.9

Line 269 is the “CT” genotype the same CT used for imaging? Please clarify.

Line 279-280 Clarify sentence.

Reviewer #2: The manuscript by Nazeer et al., “Upregulation of interleukin 1β and interleukin 18 in traumatic brain injury patients and their potential as biomarkers,” examines the elevation of IL 1β and IL 18 in TBI patients and evaluates their potential as neuroinflammatory biomarkers. In this study, serum IL 1β and IL 18 levels were quantified in samples collected from TBI patients and healthy individuals using ELISA. The authors also applied Spearman’s rank order correlation to assess the relationship between Glasgow Coma Scale scores and cytokine levels, with the goal of evaluating IL 1β and IL 18 as reliable biomarkers. The results show a negative correlation between GCS scores and IL 1β/IL 18 levels, with markedly elevated cytokine concentrations in TBI patients, supporting their potential as diagnostic and prognostic biomarkers. However, there are a couple of concerns regarding the study population that should be addressed before this manuscript would be of significant interest to the PLOS ONE readership.

1. While the demographics of the selected subjects are described, the manuscript does not provide information on patients’ medical histories, including comorbidities or pre‑existing conditions or prior concussions. These factors could significantly influence circulating IL‑1β and IL‑18 levels and should be addressed.

2. The manuscript does not indicate whether IL 1β and IL 18 levels differed between male and female TBI participants.

**Do you want your identity to be public for this peer review?** For information about this choice, including consent withdrawal, please see our For information about this choice, including consent withdrawal, please see our Privacy Policy .

Reviewer #1: No

Reviewer #2: No

---

## [Author Response · Author response to Decision Letter 1]

9 Feb 2026

We thank and appreciate the reviewers for their valuable comments. It will definitely help us to improve our manuscript. We have tried to address all the points raised by them.

Reviewer #1:

Comment: There is no justification for the choice of cytokines studied.

Response:

We thank the reviewer for this comment. IL-1β and IL-18 were specifically selected because they are key upstream mediators of post-traumatic neuroinflammation. We have added the statement to the revised manuscript, which reads as “In this work, IL-1 β and IL-18 are selected, as they are major upstream regulators of TBI-induced inflammation and secondary brain injury [27]. Unlike GFAP and UCH-L1, which are primarily upregulated as a result of neuronal and glial cell damage [28,29], these cytokines provide insight into the activation of the inflammatory pathway, aligning with the main focus of our study.”

Comment: Lines 100-105. The authors state there is no FDA-backed biomarkers for TBI. This is no longer true. Combined use of GFAP & UCH-L1 from blood is a good diagnostic tool.

Response: We thank the reviewers for highlighting this important update. We agree that our original statement was inaccurate. The combined measurement of glial fibrillary acidic protein (GFAP) and ubiquitin C-terminal hydrolase-L1 (UCH-L1) from blood has now received FDA clearance and represents a validated diagnostic tool for early assessment of traumatic brain injury, including detection of intracranial lesions after blunt head trauma. In response to this comment, we have revised the manuscript to acknowledge the FDA-approved status and clinical utility of these biomarkers. The relevant sentence has been corrected in the Introduction to show the current evidence. The modified text reads as “The pathology of traumatic brain injury is highly complex and involves many mechanisms, such as oxidative stress, neuronal damage, neuronal excitotoxicity, and inflammatory responses [24]. Although an extensive literature is present on biomarkers to assess TBI, only a few, such as the Food and Drug Administration (FDA)-approved combined measurement of glial fibrillary acidic protein (GFAP) and ubiquitin C-terminal hydrolase-L1 (UCH-L1), are currently employed for the early diagnosis of intracranial lesions following closed head injury [25]. Despite this advancement, there is continued scientific effort to identify and validate additional biomarkers linked to neuroinflammation to improve prognostic assessment, rather than serving solely as diagnostic tools [26].

Based on these considerations, we aimed to investigate the link between the key inflammatory cytokines IL-1β and IL-18 and TBI in the Pakistani population by using serum to measure their levels. Moreover, their diagnostic and prognostic potential was assessed by evaluating their AUC, sensitivity, and specificity values. In this work, IL-1 β and IL-18 are selected, as they are major upstream regulators of TBI-induced inflammation and secondary brain injury [27]. Unlike GFAP and UCH-L1, which are primarily upregulated as a result of neuronal and glial cell damage [28,29], these cytokines provide insight into the activation of the inflammatory pathway, aligning with the main focus of our study.”

Comment: Inclusion criteria are clear. However, Table 1 shows that the TBI and control groups were matched according to their demographic characteristics, but that was not stated in the methods.

Response: We thank the reviewer for this important observation. We agree that the matching of TBI and control groups should be stated in the Methods section. We have added the text. It reads now “The TBI and control groups were matched for demographic variables, including age, gender, marital status, education, and monthly income, to avoid confounding effects.”

Comment: The reported methods for obtaining and processing blood are standard. While blood was obtained on hospital admission from the TBI group, there is no indication when normals gave blood.

Response:

We thank the reviewer for this important comment. We agree that the timing of blood sampling in the control (normal) group should be clearly stated. We have modified the text in the Methods section. It reads as “For the extraction of serum, 5 mL of venous blood was drawn from TBI patients at the time of hospital admission and from healthy subjects at recruitment under resting conditions.”

Comment: The figures were not included in the submission. I got the figures from their posting to BioRxiv

Response: We apologize for this; it is possible that the figures were not opened due to a formatting issue. The figures have now been uploaded as separate files and are also included at the end of this response letter for your convenience.

Comment: Tables 2 & 3 are very repetitive with text and Figures.

Response: We appreciate the reviewer’s comment. Tables 2 and 3 provide detailed numerical and statistical data, while the figures illustrate overall trends. To avoid redundancy, the Results section has been revised and repetitive numerical descriptions have been removed. The text now focuses only on key findings.

Comment: Line 174. “negative trend”. No trend analysis was done. Negative tendency might be a better term.

Response: We agree with the reviewer. We have deleted the following sentence “A negative trend was observed in both plots, describing lower GCS scores correspond to higher IL levels.” to avoid redundancy, as it is already in (Table 2) description.

We have also analyzed the correlation between IL-1β levels and IL-18 levels. We have added results in (Table 2) and described in the Results section. The text reads as “A significant negative correlation was found between GCS score and both IL-1β (ρ = –0.430, p = 0.006) and IL-18 (ρ = -0.410, p = 0.009), demonstrating that lower GCS scores were moderately associated with higher levels of cytokines. Moreover, a moderate positive correlation was observed between IL-1β and IL-18 levels (ρ = 0.436, p = 0.005), showing that both cytokines increased together following TBI (Table 2). The scatterplots in Fig 1A-C demonstrate these correlations.”

Comment: Line 175. They should define what they consider a normal range.

Response: We thank the reviewer for this important observation. We have clarified the Results section to objectively define the normal range using values observed in healthy controls. We have modified the statement. The new text reads as “Most of the TBI patients exhibited IL-1β levels and IL-18 levels within or slightly above the range observed in healthy controls (IL-1β: 786.80–2250.70 pg/mL; IL-18: 13.78–27.91 ng/L). However, some TBI patients showed elevated cytokine levels, with IL-1β peaking at 5037.90 pg/mL and IL-18 at 85.69 ng/L.”

Comment: Line 176. The authors appropriately define what the different GCS scores mean and state that their subjects have mild TBI. However, they report a mean score of 11.05, which is indicative of moderate severity TBI. Please clarify.

Response:

We thank the reviewer for pointing this out. Our GCS data indicate that the cohort includes predominantly moderate TBI, with some patients in the mild and severe ranges. We have revised the manuscript. The modified text reads as “The GCS scores of TBI patients ranged from 6 to 14, with a mean of 11.05 ± 1.68, indicating that the cohort predominantly consisted of patients with mild-to-moderate TBI.”

Comment: Table 2. The correlation and p values for IL-1β & IL-18 are exactly the same. Is this correct?

Response: We appreciate the reviewer for his keen observations. We have analyzed the data once again and corrected the mistake. Moreover, we have added a comparison between IL-1β and IL-18. The new table appears as with modified title and legend:

“Table 2. Spearman’s rank-order correlation coefficients (ρ) for correlations between GCS score and IL-1β and IL-18 levels in TBI patients.

Association Spearman’s ρ (rho) P-value (two-tailed)

GCS versus IL-1β -0.430 0.006

GCS versus IL-18 -0.410 0.009

IL-1β versus IL-18 0.436 0.005

Positive and negative ρ values indicate positive and negative correlations, respectively. All correlations were statistically significant (p ≤ 0.05).

Abbreviations: GCS, Glasgow coma scale; IL, interleukin; TBI, traumatic brain injury.”

Comment: Line 186 & 197 is p= 0.000 a typo?

Response: Thank you for highlighting this point. The p-values reported as p = 0.000 in Lines 186 and 197 were due to statistical software rounding. These have been corrected to “p < 0.001” in the manuscript.

Comment: Line 206. The authors make a general statement about pro-inflammatory biomarkers being used for diagnosis/prognosis. Which ones.

Response:

We thank the reviewer for this important comment. In the revised manuscript, we have specified representative pro-inflammatory biomarkers that have been reported to aid in the diagnosis and prognosis of traumatic brain injury. The revised text reads as “Several pro-inflammatory biomarkers, including tumor necrosis factor (TNF)-α, apoptosis-associated speck-like protein containing a caspase recruitment domain (ASC), IL-1, IL-6, and IL-18, are described as clinically relevant biomarkers that may assist in the diagnosis and prognosis of brain trauma [33,34].”

Comment: Line 228 type. .8-9.0 should be .8-.9

Response: We have corrected it. It reads now “0.8-0.9”.

Comment: Line 269 is the “CT” genotype the same CT used for imaging? Please clarify.

Response: The CT genotype was found to be associated with an increased PTE susceptibility. In the CT genotype group

We appreciate the reviewer for pointing out this ambiguity. In the present study, the term “CT genotype” refers to the heterozygous cytosine/thymine (C/T) genotype of the analyzed single nucleotide polymorphism and is not related to computed tomography imaging or Chlamydia trachomatis. To avoid confusion, this has been clarified in the revised manuscript by defining the genotype and using “C/T”. The new text reads as “The C/T (cytosine/thymine) heterozygous genotype was found to be associated with increased susceptibility to post-traumatic epilepsy.”

Comment: Line 279-280 Clarify sentence.

Response: We have corrected the sentence. Now it reads as “as they were collected from one small part of the country.”

Reviewer #2:

Comment: 1. While the demographics of the selected subjects are described, the manuscript does not provide information on patients’ medical histories, including comorbidities or pre existing conditions or prior concussions. These factors could significantly influence circulating IL 1β and IL 18 levels and should be addressed.

Response: We thank the reviewer for raising this important concern. Potential confounding factors, including comorbidities and prior concussions, were addressed during subject enrollment. Subjects with pre-existing inflammatory, autoimmune, infectious, or neurological disorders, as well as those with a prior history of TBI or concussion, were not recruited for this study. We have now stated this exclusion criterion in the Methods section, which reads as “Patients with pre-existing inflammatory, autoimmune, infectious, or neurological disorders, as well as those with a prior history of TBI or concussion, were excluded from this study to avoid confounding effects on circulating cytokine levels.”

Comment: 2. The manuscript does not indicate whether IL 1β and IL 18 levels differed between male and female TBI participants.

Response: We thank the reviewer for this valuable comment. We have analyzed IL-1β and IL-18 levels according to gender. These results are added to the Results section with (Table 3) and (Fig 4). The text reads as “Levels of cytokines were also analyzed to assess their distribution between male and female participants. IL-1β levels were not significantly different between males and females of normal (p = 0.15) and TBI groups (p = 0.352). On the contrary, in the normal group, females had slightly but significantly higher levels of IL-18 than males (19.93 ± 2.64 vs. 18.67 ± 3.14 ng/L; p = 0.043). In TBI subjects, this difference was significantly higher (p = 0.012) in females (28.91 ± 18.03) as compared to males (23.48 ± 11.41 ng/L) (Table 3, Fig 3).”

Table 3. Gender-based comparison of cytokines in normal and TBI subjects.

Cytokine Group Male (Mean ± SD) Female (Mean ± SD) P-value

IL-1β

Normal 1538.05 ± 279.87 1365.92 ± 393.54 0.15

TBI 1907.47 ± 760.48 1851.29 ± 860.21 0.352

IL-18 Normal 18.67 ± 3.14 19.93 ± 2.64 0.043

TBI 23.48 ± 11.41 28.91 ± 18.03 0.012

Abbreviations: IL, interleukin; TBI, traumatic brain injury; SD, standard deviation

Fig 3. Gender-based comparison of IL-1β (A) and IL-18 (B) levels in normal and TBI participants. Data are presented as mean ± standard deviation. Statistical significance was assessed using the Mann–Whitney U test. IL-1β levels did not differ significantly between sexes in either group (p > 0.05), whereas IL-18 levels were significantly higher in females than in males in both the normal and TBI groups.

We have also added the following paragraph to the Discussion chapter “Our data demonstrate a differential post-TBI inflammatory response between male and female subjects, denoted by higher IL-1β levels in males and elevated IL-18 levels in females. Our results align with the previous research work that also demonstrates gender-specific inflammatory response following TBI. For instance, in an animal study [48], male mice exhibited a significant increase in IL-1β levels compared to females following experimental TBI. Conversely, gender-specific data on IL-18 levels in the context of TBI are sparse. However, our findings are supported by a previous study [49] in which higher serum levels of IL-8, another related inflammatory cytokine, were demonstrated in TBI females compared to males. Taken together, our data demonstrate that females may exhibit a more robust cytokine response following TBI, which highlights the importance of gender-specific considerations in TBI management and immune-targeted therapeutic strategies.

In addition to addressing the reviewers’ suggestions, we have made minor corrections throughout the manuscript, which are visible in the tracked-changes manuscript.

Fig 1. Scatterplots exhibiting association between Glasgow coma scale (GCS) scores and serum interleukin (IL) levels. GCS scores were significantly negatively correlated (Spearman’s ρ = –0.43, p = 0.006) with IL-1β (A) and IL-18 (B) in TBI patients.

Fig 2. Box plots exhibiting comparative analysis of interleukin (IL) levels in serum of traumatic brain injury (TBI) patients (n = 40) and normal individuals (n = 40), as assessed by enzyme-linked immunosorbent assay (ELISA). Levels of (A) IL-1β and (B) IL-18 are higher in TBI patients compared with normal individuals. Each box indicates interquartile range (IQR), the horizontal line inside the box exhibits the median, whiskers depict variability outside the upper and lower quartiles, circles (°) denote mild outliers (values between 1.5 and 3 times the IQR), and asterisks (*) indicate extreme outliers (values exceeding 3 times the IQR). Statistical significance was assessed using the Mann–Whitney U-test.

Fig 3. Gender-based comparison of IL-1β (A) and IL-18 (B) levels in normal and TBI participants. Data are presented as mean ± standard deviation. Statistical significance was assessed using the Mann–Whitney U test. IL-1β levels did not differ significantly between sexes in either group (p > 0.05), whereas IL-18 levels were significantly higher in females than in males in both the normal and TBI groups.

Fig 4. The area under the receiver operating characteristic (ROC) curve (AUC) and the discriminatory ability of IL-1β and IL-18. The black line in the figure represents IL-1β, and the grey line denotes IL-18. The Y-axis shows the ability of ILs to correctly detect TBI patients, and the X-axis represents the proportion of healthy individuals wrongly classified as TBI. The higher AUC value of IL-18 (0.78) than that of IL-1β (0.72) demonstrates its better diagnostic power as a biomarker.

---

## [Decision Letter · Decision Letter 1]

12 Mar 2026

Upregulation of interleukin-1β and interleukin-18 in traumatic brain injury patients and their potential as biomarkers

PONE-D-25-55326R1

Dear Dr. Murtaza,

We’re pleased to inform you that your manuscript has been judged scientifically suitable for publication and will be formally accepted for publication once it meets all outstanding technical requirements.

Kind regards,

Subhra Mohapatra, MS PhD

Academic Editor

PLOS One

Additional Editor Comments (optional):

Reviewers' comments:

Reviewer's Responses to Questions

**Comments to the Author**

Reviewer #1: All comments have been addressed

Reviewer #2: All comments have been addressed

2. Is the manuscript technically sound, and do the data support the conclusions?

Reviewer #1: Yes

Reviewer #2: Yes

3. Has the statistical analysis been performed appropriately and rigorously?

Reviewer #1: Yes

Reviewer #2: Yes

4. Have the authors made all data underlying the findings in their manuscript fully available?

Reviewer #1: Yes

Reviewer #2: Yes

5. Is the manuscript presented in an intelligible fashion and written in standard English?

Reviewer #1: Yes

Reviewer #2: Yes

Reviewer #1: (No Response)

Reviewer #2: (No Response)

**Do you want your identity to be public for this peer review?** For information about this choice, including consent withdrawal, please see our For information about this choice, including consent withdrawal, please see our Privacy Policy .

Reviewer #1: No

Reviewer #2: No

---

## [Editor Report · Acceptance letter]

PONE-D-25-55326R1

PLOS One

Dear Dr. Murtaza,

I'm pleased to inform you that your manuscript has been deemed suitable for publication in PLOS One. Congratulations! Your manuscript is now being handed over to our production team.

Kind regards,

on behalf of

Dr. Subhra Mohapatra

Academic Editor

PLOS One